# DUET: JOINT EXPLORATION OF USER–ITEM PROFILES

## ABSTRACT

Traditional recommendation systems represent users and items as hidden vectors, learning to align them in a shared latent space for relevance estimation. With the advent of large language models (LLMs), we advocate a shift *from vectors to text*: representing users and items as *textual profiles* and aligning them in a shared semantic space. Textual profiles are directly compatible with LLMs and offer interpretability for downstream agentic systems. A key challenge, however, is that the optimal profile format is unknown, and handcrafted templates often misalign with task objectives. We propose DUET, a framework for *joint exploration of user–item profile generation in text*. The framework operates in three stages. The framework operates in three stages. First, raw histories and metadata are distilled into simple *cues* that capture minimal but informative signals. Second, during a single sequence-to-sequence inference pass, these cues are expanded into richer prompts and then into textual profiles, allowing for the exploration of multiple formats rather than a single creation. Finally, profiles are optimized jointly via reinforcement learning, where downstream recommendation performance provides feedback to refine and align them. Experiments on three real-world datasets demonstrate that DUET outperforms strong baselines, validating the effectiveness of joint textual profile alignment and the utility of prompt-driven exploration. Project page: https://duetreview.github.io/.

## 1 INTRODUCTION

Traditional recommendation systems represent users and items as hidden vectors, learning to align them in a shared latent space for relevance estimation (Covington et al., 2016; Wu, 2023).
While effective, such vector embeddings are opaque, difficult to adapt, and fundamentally mismatched with the text-based reasoning style of large language models (LLMs). With the advent of LLMs, a shift is needed *from vectors to text*: representing users and items as *textual profiles* and aligning them in a shared semantic space. Textual profiles are directly compatible with LLMs, naturally support downstream agentic recommendation systems, and provide interpretability while flexibly integrating heterogeneous signals such as histories, metadata, and reviews. This shift preserves the alignment principle at the heart of recommendation while opening new opportunities for LLM-driven personalization.

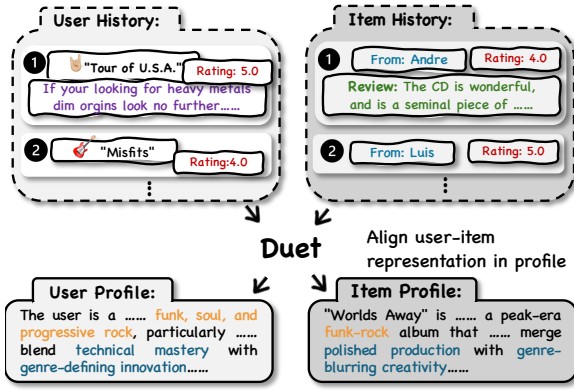

Figure 1: DUET aligns raw user and item data by transforming them into textual profiles within a shared semantic space.

The motivation for textual alignment is straightforward. User and item data are inherently heterogeneous: a user is described by past interactions, while an item is characterized by attributes such as price, category, or reviews. Represented independently, these signals are difficult to compare. By converting both into textual profiles within a shared semantic space, their relationship becomes interpretable. For example, as shown in Fig. 1, a user profile may describe "enjoys strategy-focused

puzzle games," while an item profile may describe "an indie puzzle game for players seeking challenging problem-solving experiences." This textual alignment enables direct comparison and reasoning over user–item compatibility.

Recent work has begun leveraging LLMs to enhance recommendation by generating semantically rich and interpretable representations (Wang et al., 2025; Zhang, 2024; Kronhardt et al., 2024; Purificato et al., 2024; Xi et al., 2024; Bao et al., 2023). However, existing approaches remain limited. Directly feeding raw user histories into LLMs often yields noisy and incomplete signals, while template-based profiles require extensive human engineering and fail to capture the evolving complexity of user preferences and item semantics (Yang et al., 2023; Xi et al., 2024; Wang et al., 2025). A central challenge, therefore, is that the optimal textual profile format is unknown, and handcrafted templates often misalign with downstream objectives.

To address this gap, this work introduces DUET, a framework for *joint exploration of user–item profile generation in text*. DUET operates in three stages. First, raw histories and metadata are distilled into simple *cues* that capture minimal but informative signals. Second, in the same seq-to-seq inference pass, these cues are expanded into richer prompts and then into textual profiles—enabling exploration of alternative formats rather than one-shot creation. Finally, profiles are optimized jointly via reinforcement learning, where downstream recommendation performance provides feedback to refine and align them. By coupling both sides of the interaction in a shared semantic space, DUET learns user profiles that capture what kinds of items a user prefers and item profiles that capture what kinds of users an item appeals to.

The contributions of this work are threefold: (1) Representing users and items as textual profiles and aligning them in a shared semantic space, extending the classic vector-based alignment principle to natural language representations. This shift establishes a foundation for future agentic recommendation systems. (2) An exploration-based framework is proposed that begins with cue-based initialization, expands cues into richer prompts and profiles through re-prompt generation, and jointly optimizes user and item profiles with downstream recommendation feedback. This design enables the LLM to autonomously discover effective profile formats without rigid templates. (3) Extensive experiments across multiple real-world datasets demonstrate that DUET consistently outperforms strong baselines, validating both the effectiveness of textual alignment and the utility of feedback-driven profile exploration.

## 2 RELATED WORK

### 2.1 PROFILES IN RECOMMENDATION

Early recommendation systems primarily relied on pre-defined profiles based on structured attributes, as seen in works like CRESDUP Chen et al. (2007) and UPCSim Widiyaningtyas et al. (2021). While foundational, these methods were limited by their rigid, hand-engineered features. More recently, the advent of Large Language Models (LLMs) has enabled a shift towards generating profiles in natural language. Studies such as KAR Xi et al. (2024), GPG Zhang (2024), PALR Yang et al. (2023), and LettingGo Wang et al. (2025) leverage LLMs to create textual user profiles from behavioral data. A key limitation of these approaches is twofold: they rely on static or **pre-defined templates** for profile generation, and they often focus exclusively on user profiles, neglecting the rich, expressive information inherent in items and the complex dynamics of user-item interactions. In contrast, our work introduces a new paradigm where both user and item profiles are not fixed but are dynamically explored and optimized in a shared semantic space to directly align with recommendation performance.

### 2.2 REINFORCEMENT LEARNING FOR LLM-BASED RECOMMENDATION SYSTEMS

Reinforcement Learning (RL), particularly through techniques like RLHF, has become a core method for aligning LLMs with specific objectives. This approach has been adapted for recommendation tasks Wang et al. (2025); Lin et al. (2025); Deng et al. (2025), but existing efforts often face key limitations. They frequently rely on offline reward models (Jeong et al., 2023; Chen et al., 2025) that do not adapt in real-time to system feedback, a setup that risks issues like reward hacking (Skalse et al., 2025). Other methods (Sun et al., 2024; Lu et al., 2024) restrict themselves to offline preference tuning (e.g., DPO), which can easily overfit on static datasets. Our framework,

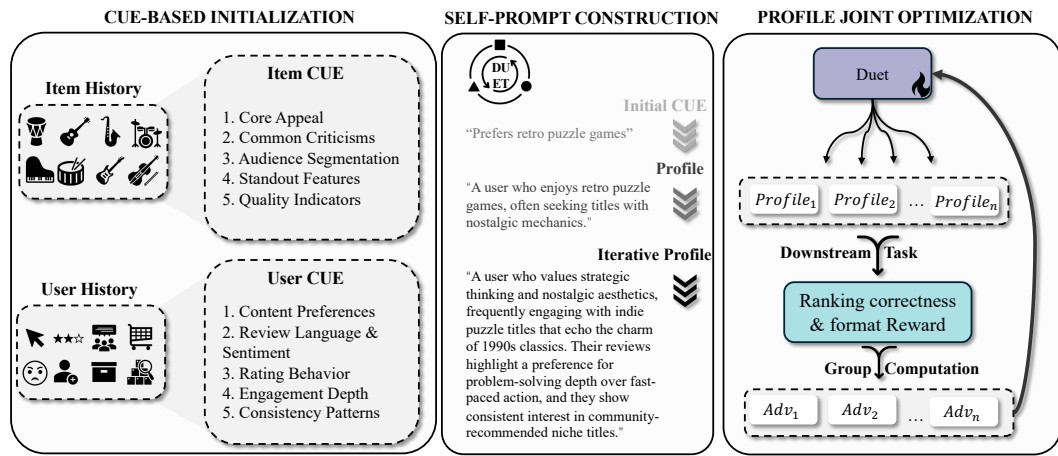

Figure 2: Overview of the DUET framework. It operates in three stages: cue-based initialization of user and item histories, self-prompt construction to explore richer textual profiles, and joint optimization with downstream feedback to align profiles in a shared semantic space.

DUET, overcomes these challenges by integrating RL into a closed-loop system where downstream recommendation performance serves as the real-time reward signal, allowing for the dynamic and interactive refinement of textual profiles.

## 3 METHOD

As illustrated in Fig. 2, DUET is a closed-loop framework for *exploring user–item profile generation*. The process unfolds in three components. First, **cue-based initialization** distills raw user histories and item metadata into concise textual cues that provide minimal but informative signals. Second, **self-prompt construction** performs exploration in a single seq-to-seq pass: each cue is rewritten into a richer prompt, and within the same generation DUET produces the corresponding textual profile. Finally, **joint profile optimization** closes the loop by aligning user and item profiles with downstream recommendation feedback via reinforcement learning, ensuring that the resulting representations are both semantically coherent and performance-aligned. In this way, DUET is able to explore and adapt the optimal prompt formats for textual profile generation

### 3.1 PROBLEM FORMULATION

Let $\mathcal{U}$ denote the set of users and $\mathcal{I}$ the set of items. Each user $u \in \mathcal{U}$ is associated with an interaction history $H_u = \{i_1, i_2, \ldots, i_m\}$, and each item $i \in \mathcal{I}$ has metadata $M_i$ such as category, attributes, or textual descriptions. The objective is to generate textual profiles $P_u$ and $P_i$ that capture salient properties of users and items, respectively, in a shared semantic space.

In the downstream recommendation task, a scoring function $f(P_u, P_i \mid H_u^{(k)})$ estimates the relevance $\hat{y}_{ui}$ of item $i$ for user $u$, where $H_u^{(k)}$ denotes the most recent $k$ interactions (e.g., the last 10 items clicked or consumed). This formulation mirrors traditional recommendation systems, where latent vectors $\mathbf{u}$ and $\mathbf{i}$ are aligned in an embedding space. The key difference is that DUET replaces opaque vectors with natural language descriptions, enabling profiles to serve as both semantically rich inputs to LLM-based recommenders and interpretable signals for downstream agentic systems.

### 3.2 CUE-BASED INITIALIZATION

Raw user histories and item metadata are often noisy, redundant, and not directly suitable for profile construction. To address this, DUET introduces the concept of *cues*: concise hypotheses that summarize minimal but informative aspects of users and items. These cues act as lightweight seeds, which are deliberately underspecified so that the system can subsequently explore richer profile formats.

**Definition 1** (Cue). A *cue* is a minimal textual hypothesis derived from historical data that highlights one potential aspect of a user's preference or an item's characteristic. Rather than aiming for completeness, cues capture partial but salient signals that serve as starting points for profile exploration.

To extract cues automatically, the LLM is prompted to summarize minimal but informative aspects of user or item data. For example, given a user's interaction history, the model is guided with instructions such as:

> **Cue Extraction Prompt**
>
> *"From the history below, analyze the user's historical interactions to understand preferences, rating behavior, review sentiment or any other dimension. Keep the description concise and avoid full sentences."*

This lightweight guidance allows the LLM to map raw histories and metadata into compact textual cues. The following examples illustrate how raw signals are distilled into cues. As shown on the top, user cues emphasize historical preferences, while item cues highlight metadata and user-group patterns. Together, they provide minimal but informative hypotheses for profile exploration.

> **Examples of User Cues**
> *"enjoys retro puzzle games"* — derived from repeated engagement with classic titles.
> *"prefers concise product reviews"* — inferred from a pattern of short, direct comments.
> *"tends to give high ratings but rarely comments"* — highlighting consistency but limited feedback.

> **Examples of Item Cues**
> *"lightweight trail-running shoes"* — derived from product metadata.
> *"popular among budget-conscious users"* — inferred from purchase patterns.
> *"stylized with retro aesthetics"* — extracted from item descriptions.

In summary, cue-based initialization provides DUET with a compact, interpretable, and flexible entry point for profile generation. Rather than forcing a fixed template, cues establish minimal but meaningful hypotheses that can be expanded during exploration to discover effective profile formats.

### 3.3 EXPLORATION VIA SELF-PROMPT CONSTRUCTION

The key innovation of DUET lies in its ability to *explore* effective textual profile formats rather than fixing them a priori. While cues provide minimal but informative hypotheses, they are deliberately underspecified and cannot fully capture user preferences or item semantics. To address this, DUET employs a *self-prompt construction* mechanism: starting from cues, the LLM is re-prompted with lightweight instructions that encourage elaboration, contextualization, and refinement. This process transforms short cues into coherent and semantically rich profiles, enabling diverse exploration of possible formats in a single sequence-to-sequence inference pass.

> **Exploration Prompt(excerpt)**
>
> *"Rewrite the given cue into a detailed profile. Expand on behavioral tendencies, stylistic preferences, or characteristic features, while ensuring the description remains coherent and useful for recommendation. Keep the profile concise (1–3 sentences)."*

The following example illustrates how a simple cue is expanded into a richer profile through the exploration step:

> **Exploration Example(excerpt)**
>
> **Cue:** *"enjoys fantasy novels"*
> **self-prompt construction:** *"Based on the cue, generate a profile that focuses on the user's preferred sub-genres, typical themes, and reading habits."*
> **Expanded Profile:** *"A reader who is drawn to high fantasy, often preferring intricate world-building and epic narratives."*

In this way, DUET avoids reliance on rigid templates and instead *explores* alternative profile formulations. The re-prompting step encourages the model to discover richer descriptions that better align with downstream recommendation objectives, ultimately producing profiles that are both interpretable and performance-oriented.

In practice, the first two stages of DUET—cue extraction and self-prompt construction—are realized within a single seq-to-seq generation pass. The model is prompted with raw histories to produce, in sequence, a concise cue, an expanded prompt, and the final textual profile. This design avoids multi-step inference while still enabling exploration of richer profile formats. Figure 3 illustrates how cue extraction and self-prompt construction are realized as a single sequence-to-sequence generation process.

**Seq-to-Seq Input**
**Cue Extraction Prompt (excerpt):** *"From the history below, extract one short phrase that best captures the user's tendency or preference. Keep it concise; no full sentences."*
**Re-prompt for Profile Construction (excerpt):** *"Using the cue, construct a profile prompt that specifies what to describe (e.g., behavioral tendencies, stylistic preferences, engagement depth). The profile will be 1–3 sentences in natural language."*

**Seq-to-Seq Output**
**Cue:** *"prefers retro puzzle games"*
**Constructed Profile Prompt:** *"Generate a concise user profile focusing on strategic challenge, nostalgic aesthetics, and typical engagement depth."*
**Profile:** *"A user who values strategic thinking and nostalgic aesthetics, frequently engaging with indie puzzle titles reminiscent of 1990s classics."*

Figure 3: Implementation of cue extraction and self-prompt construction as a single sequence-to-sequence generation process in DUET.

### 3.4 DUET: USER–ITEM PROFILE JOINT OPTIMIZATION

A central goal of DUET is to ensure that user and item profiles are not optimized in isolation, but jointly aligned in a shared semantic space. We therefore formulate profile generation as a reinforcement learning (RL) problem, where the profile generator $f_{\text{LLM}}$ takes cues $(C_u, C_i)$ as input and produces textual profiles $(P_u, P_i)$. The downstream recommendation task acts as the environment: a scoring function $f(P_u, P_i)$ predicts relevance $\hat{y}_{ui}$, and observed interactions provide feedback as rewards. This design naturally couples user and item profiles, since the reward reflects their mutual alignment. We optimize $f_{\text{LLM}}$ with Group Relative Policy Optimization (GRPO) (DeepSeek-AI et al., 2025).

**Reward Design.** A critical challenge arises because recommendation labels (e.g., 1–5 ratings) are inherently discrete. If treated as strict integer targets, the reward surface becomes overly coarse: predicting a rating of 4 when the ground truth is 5 receives the same penalty as predicting 2, despite being much closer in intent. Such coarse signals discourage the model from refining subtle profile details and often lead to conservative strategies. For ranking-oriented recommendation, however, systems naturally output *continuous scores*, which better capture fine-grained preferences.

To reflect this, we define the performance reward as a continuous function of the distance between the predicted score $\hat{y}_{ui}$ and the ground-truth label $y_{ui}$:

**Definition 2** (Fractional Performance Reward). For a user $u$ and item $i$ with ground-truth label $y_{ui} \in \{1, 2, 3, 4, 5\}$ and predicted score $\hat{y}_{ui} \in \mathbb{R}$, the fractional reward is defined as:

$$R_{\text{perf}}(u, i) = 1 - \frac{|y_{ui} - \hat{y}_{ui}|}{M},$$

where $M$ is the maximum rating gap (e.g., $M = 4$ for a 1–5 scale).

This formulation ensures that predictions closer to the ground truth receive proportionally higher rewards. For example, if $y_{ui} = 5$, then $\hat{y}_{ui} = 4.8$ yields $R_{\text{perf}} = 0.95$, whereas $\hat{y}_{ui} = 3.0$ yields $R_{\text{perf}} = 0.50$. Thus, unlike integer-based rewards that penalize both equally, fractional rewards provide smooth gradients.

## 4 EXPERIMENT

In this section, we present the experimental evaluation results of our framework across multiple datasets, addressing the following research questions:

- RQ1: How does DUET compare to traditional fixed-format profiles in recommendation systems?
- RQ2: How does feedback-driven alignment improve profile generation quality?
- RQ3: How does historical interaction length affect profile quality and task performance?
- RQ4: How does DUET enhance interpretability and capture diverse user preferences?

### 4.1 EXPERIMENTAL SETTINGS

#### 4.1.1 DATASETS.

Experiments were conducted on three widely used real-world datasets. **Amazon Music (Music)** and **Amazon Book (Book)** are derived from the Amazon Product dataset[1], while **Yelp** is from the Yelp Open dataset[2]. All datasets include user reviews, ratings, and rich textual information. We used the full Amazon Music dataset, but only subsets (latest two months for Book and six for Yelp). Data was split by timestamp into training, validation, and test sets to prevent information leakage Ji et al. (2023). We then applied 5-core filtering Liu et al. (2019) and removed cold-start users and items from the validation and test sets. Statistical details of the processed datasets are provided in Table 4.

#### 4.1.2 EVALUATION METRICS.

Following the experimental paradigm of rating prediction Fang et al. (2025), we evaluate the effectiveness of generated user profiles by directly leveraging LLMs and textual data, similar to GPG Zhang (2024) and the approach in Kang et al. (2023). For each user, the downstream recommendation system predicts a rating for a candidate item based on three inputs: the user's recent history, a generated user profile, and a generated item profile. The objective is to assess how well these profiles support accurate predictions. A correct classification validates the profile as effective, whereas an incorrect one suggests it is less representative of the user's preferences or the item's appeals. We employ four widely-adopted standard metrics Wang et al. (2025); Fang et al. (2025): **Mean Absolute Error (MAE)**, **Root Mean Square Error (RMSE)**, **Accuracy**, and **F1 score**.

#### 4.1.3 BASELINES.

To enable a comprehensive comparison and demonstrate the effectiveness of our proposed approach, we implement several key baseline methods. The most straightforward are the direct prediction methods, including **10 Recent History (10H)**, which uses the 10 most recent interactions, and **KAR** Xi et al. (2024), which also uses a fixed number of interactions but acknowledges that long histories can be noisy. In contrast, other approaches leverage large language models (LLMs) to first generate user profiles. We include **RLMRec** Ren et al. (2024) and **PALR** Yang et al. (2023), both of which prompt LLMs to create recommendation-focused profiles from user data, differing mainly in their

---

[1] https://cseweb.ucsd.edu/~jmcauley/datasets/amazon/links.html
[2] https://business.yelp.com/data/resources/open-dataset/

Table 1: Performance on three datasets using Qwen 3 (8B) and LLaMA 3 (8B).

| Method | Yelp | | | | Amazon Music | | | | Amazon Books | | | |
|---|---|---|---|---|---|---|---|---|---|---|---|---|
| | MAE | RMSE | Acc (%) | F1 (%) | MAE | RMSE | Acc (%) | F1 (%) | MAE | RMSE | Acc (%) | F1 (%) |
| Qwen 3 (8B) | | | | | | | | | | | | |
| 10H | 1.1235 | 1.9478 | 23.17 | 27.54 | 0.9102 | 1.4021 | 39.26 | 46.58 | 0.9314 | 1.4527 | 37.63 | 45.19 |
| KARXi et al. (2024) | 0.7396 | 1.2184 | 55.34 | 48.67 | 0.7483 | 1.1380 | 58.65 | 60.29 | 0.7098 | 1.0923 | 56.17 | 58.78 |
| RLMRecRen et al. (2024) | 0.8197 | 1.3312 | 47.15 | 42.46 | 0.7438 | 1.1069 | 54.89 | 57.65 | 0.7812 | 1.1584 | 52.86 | 55.93 |
| PALRYang et al. (2023) | 0.7994 | 1.2876 | 48.53 | 43.19 | 0.6075 | 0.9531 | 57.35 | 56.77 | 0.7485 | 1.1187 | 54.24 | 56.38 |
| LGWang et al. (2025) | 0.6632 | 1.1047 | 56.18 | 48.95 | 0.4737 | 0.8834 | 62.37 | 57.09 | 0.5821 | 0.9416 | 59.35 | 60.57 |
| R4RecFang et al. (2025) | 0.7028 | 1.1523 | 55.69 | 47.73 | 0.5654 | 0.9635 | 58.69 | 54.67 | 0.6397 | 1.0098 | 58.47 | 56.84 |
| **Ours** | **0.5126** | **0.9485** | **61.23** | **55.18** | **0.3937** | **0.7564** | **67.96** | **63.89** | **0.4612** | **0.9089** | **64.38** | **59.27** |
| LLaMA 3 (8B) | | | | | | | | | | | | |
| 10H | 1.0864 | 1.9532 | 22.09 | 27.30 | 0.7917 | 1.3346 | 38.13 | 46.87 | 0.8064 | 1.3866 | 37.15 | 45.27 |
| KARXi et al. (2024) | 0.6427 | 1.1668 | 54.51 | 47.98 | 0.5726 | 0.9033 | 57.53 | 59.92 | 0.5892 | 0.9614 | 55.87 | 58.21 |
| RLMRecRen et al. (2024) | 0.7428 | 1.3572 | 46.74 | 42.11 | 0.6076 | 0.9886 | 53.78 | 57.42 | 0.6226 | 0.9477 | 52.12 | 55.79 |
| PALRYang et al. (2023) | 0.7238 | 1.3265 | 47.72 | 43.29 | 0.5823 | 0.9222 | 56.73 | 59.31 | 0.5977 | 0.8855 | 55.06 | 57.62 |
| LGWang et al. (2025) | 0.6196 | 1.1289 | 56.03 | 51.24 | 0.5204 | 0.9369 | 61.92 | 59.50 | 0.5543 | 0.7967 | 58.95 | 60.39 |
| R4RecFang et al. (2025) | 0.7586 | 1.0418 | 55.80 | 53.00 | 0.5442 | 0.7722 | 60.86 | 54.88 | 0.6029 | 0.8345 | 59.70 | 56.35 |
| **Ours** | **0.5367** | **0.9687** | **60.87** | **54.74** | **0.4680** | **0.8277** | **63.30** | **60.60** | **0.5092** | **0.9500** | **63.42** | **58.12** |

specific prompts. More advanced frameworks are also evaluated. **LG (LettinGo)** Wang et al. (2025) uses a multi-stage process alignment to build adaptive, diverse profiles, while **Reason4Rec** Fang et al. (2025) trains a collaborative three-expert system to reason before predicting, which improves accuracy and reduces data bias.

## 4.2 MAIN RESULTS (RQ1)

Table 1 presents a comparison of our proposed method against five state-of-the-art baselines on three datasets: Amazon Music, Amazon Books, and Yelp. We use Qwen3-8B (Team, 2025) and LLaMA3-8B (Dubey et al., 2024) as both the profile generator and the downstream recommendation model, with a prediction temperature of 0. The experimental results demonstrate that our approach achieves significant performance improvements across all datasets. On the Qwen3-8B model, our method improves accuracy by an average of 30 percentage points over the baseline that uses only historical interactions. Compared to all other profile- and reasoning-based approaches, our method exhibits a clear advantage, achieving an average accuracy of 64.52% and an F1 score of 60.80%. This performance superiority can be attributed to three key factors. First, our method generates **structured and informative profiles** through carefully designed prompt templates, which effectively capture core user preferences. Second, it employs **dynamic and balanced preference modeling** by combining the most recent 10 interactions with the generated profile information, which balances short-term and long-term user interests. Finally, through **profile exploration and optimization**, our framework leverages downstream feedback to align the profile generator with the recommendation task, consistently producing more effective profiles. We also observed differences in performance improvements across datasets, with the largest relative improvement on the Amazon Music dataset, suggesting that long-term user interests have a more significant impact in this domain.

## 4.3 ABLATION STUDY

### 4.3.1 EFFECTIVENESS OF ALIGNMENT (RQ2)

To investigate the impact of profile alignment on the final performance, we conducted ablation experiments on three datasets using the same experimental settings described in Section 4.1 on the Qwen3 model. Specifically, we compared the performance before and after optimizing with GRPO (Group Relative Policy Optimization ). This analysis helps us understand how alignment contributes to generating more task-relevant and high-quality profiles. As shown in Table 2, our profile alignment method yields consistent and substantial improvements in accuracy across all three datasets. On the Yelp dataset, it improves accuracy by 5.4%. The effect is even more pronounced on the

Table 2: An ablation study on each design in DUET (using Qwen 3 (8B)).

| Method | Yelp | | | | Amazon Music | | | | Amazon Books | | | |
|---|---|---|---|---|---|---|---|---|---|---|---|---|
| | MAE | RMSE | Acc (%) | F1 (%) | MAE | RMSE | Acc (%) | F1 (%) | MAE | RMSE | Acc (%) | F1 (%) |
| 10H | 1.1235 | 1.9478 | 23.17 | 27.54 | 0.9102 | 1.4021 | 39.26 | 46.58 | 0.9314 | 1.4527 | 37.63 | 45.19 |
| +Profile | 0.7218 | 1.1863 | 55.48 | 48.09 | 0.6597 | 1.0218 | 58.67 | 57.48 | 0.6764 | 1.0469 | 57.14 | 57.68 |
| +Self - Prompt | 0.7085 | 1.1654 | 55.83 | 48.54 | 0.5708 | 0.9897 | 58.91 | 55.53 | 0.6389 | 1.0108 | 58.43 | 56.88 |
| +Alignment | 0.5126 | 0.9485 | 61.23 | 55.18 | 0.3937 | 0.7564 | 67.96 | 63.89 | 0.4612 | 0.9089 | 64.38 | 59.27 |

Table 3: Impact of historical interaction length on profile quality (using Qwen 3 (8B)).

| Method | Yelp | | | | Amazon Music | | | | Amazon Books | | | |
|---|---|---|---|---|---|---|---|---|---|---|---|---|
| | MAE | RMSE | Acc (%) | F1 (%) | MAE | RMSE | Acc (%) | F1 (%) | MAE | RMSE | Acc (%) | F1 (%) |
| 10H+30P | 0.5126 | 0.9485 | 61.23 | 55.18 | **0.3883** | **0.7494** | **67.96** | **63.89** | 0.4612 | 0.9089 | **65.13** | **59.97** |
| 10H+50P | **0.4909** | **0.9207** | **62.43** | **56.24** | 0.3924 | 0.7543 | 67.88 | 63.87 | **0.4553** | **0.9023** | 64.62 | 59.52 |
| 10H+70P | 0.4987 | 0.9326 | 61.98 | 55.81 | 0.3937 | 0.7564 | 68.22 | 64.12 | 0.4608 | 0.9068 | 64.38 | 59.27 |

Amazon Books dataset, with a 5.95% improvement, and reaches its peak on the Amazon Music dataset, where accuracy increases by 9.05%. These results demonstrate that profile alignment effectively leverages feedback from downstream tasks to guide the profile generation model. By learning to produce profile formats that are better aligned with task-specific objectives, the method significantly enhances the quality of generated profiles, more accurately summarizes user preferences, and ultimately improves downstream recommendation performance.

### 4.3.2 IMPACT OF HISTORICAL INTERACTION LENGTH ON PROFILE QUALITY.

We conducted experiments to evaluate the impact of historical interaction length on the quality of the generated profiles, using lengths of 30, 50, and 70 interactions. The results, summarized in Table 3, show that longer histories do not consistently produce better profiles; instead, performance varies depending on the dataset and experimental conditions. This phenomenon can be attributed to several factors. First, **Dataset Scale and Sparsity** influence the outcome: in sparse datasets, shorter histories are often sufficient, while longer histories may introduce noise. In contrast, longer histories can provide richer semantic information in dense datasets. Second, **User Interest Dynamics** play a role, as shorter histories better reflect recent and more relevant preferences, whereas longer histories may include outdated or irrelevant interactions. Finally, **Model Capacity and Robustness to Noise** are critical, as models with higher capacity can better extract meaningful signals from longer histories, while limited-capacity models may struggle with excessive input. These findings suggest that the choice of historical interaction length should be adapted dynamically, taking into account the characteristics of the dataset and the capacity of the recommendation model, which helps optimize the trade-off between capturing sufficient user preferences and avoiding noise.

### 4.4 CASE STUDY

#### 4.4.1 EFFECTIVENESS OF DUET PROFILE (RQ4).

A key research question is whether the profiles generated by large language models (LLMs) can effectively capture meaningful signals from user and item histories. As shown in Figure 4, the generated profiles indeed condense and summarize important information that is scattered across individual reviews. Moreover, since user and item profiles are jointly optimized, there exists a strong correspondence between them. This alignment enables the model to capture fine-grained preference–attribute matches, leading to more accurate score predictions.

As shown in Figure 4, this is an example where the model makes incorrect predictions when using only a limited number of historical records, but successfully predicts when additional profile

**User History:**
Andre' S Grindle "Tour of U.S.A.", Rating: 5.0. If your looking for heavy metals dim orgins look no further.
Andre' S Grindle "Misfits", Rating:4.0. Actually the opening title track is more of a direct pop/rock type song. But the rock n roll returns in a big way on "Hay Fever".
············

**Item History:**
Worlds Away, Rating: 4.0, Review: The CD is wonderful, and is a seminal piece of music for both the group and for those of us who were 'of age' in the '70s and '80s. Pablo Cruise was - and is - a group that makes beautiful music like few others, and this shipper/seller made it all the more special with a fast ship! THANKS!!!
Worlds Away, Rating: 5.0, I love this CD! I'd recommend the Pablo Cruise Millenium Collection to get to know this band. Worlds Away to me represents the best of breezy '70s pop. Five stars!
············

**USER PROFILE:**
Andre Grindle is a dedicated music enthusiast with a deep appreciation for funk, soul, and progressive rock, particularly drawn to artists who blend technical mastery with genre-defining innovation. His reviews consistently highlight albums that showcase musical complexity, lyrical depth, and historical significance, often referencing artists like Stevie Wonder, Prince, and Rush. He values authenticity and artistic vision, frequently praising albums that resist trend-chasing and prioritize creative integrity. His detailed, introspective reviews suggest a preference for nuanced analysis over superficial praise, with a focus on emotional resonance, production quality, and the artist's evolution. While he is generous with high ratings (often 5.0), he is also willing to critique perceived shortcomings, as seen in his 4.0 for "Test for Echo" and 3.0 for "The Color of Success." His engagement is characterized by a blend of personal anecdotes, technical appreciation, and historical context, indicating a preference for music that transcends mere entertainment.

**Item PROFILE:**
"Worlds Away" by Pablo Cruise is celebrated as a peak-era funk-rock album that exemplifies the band's ability to merge polished production with genre-blurring creativity. Users consistently praise its tight musicianship, genre-defying sound, and the band's ability to craft catchy, emotionally resonant tracks. The album's appeal lies in its seamless fusion of rock, funk, and pop, with standout tracks like "Love Will Find a Way" and "Worlds Away" receiving particular acclaim for their infectious melodies and lyrical depth. While some users note its dated production compared to modern standards, the majority highlight its historical significance as a defining work of the 1970s rock-funk movement. The album resonates most with fans of 1970s progressive rock, genre-defying artists, and listeners seeking well-crafted, emotionally engaging music. Its high average rating reflects widespread admiration for its artistic ambition and enduring appeal, though a minority acknowledges its limitations in terms of modern production quality.

Figure 4: Illustration of the mutual correspondence between user and item. The highlighted regions demonstrate that user preferences summarized in the user profile align with the key attributes extracted in the item profile, which provides complementary information beyond raw histories and thus improves prediction accuracy.

information is added. The colored highlights indicate the semantic correspondence between USER HISTORY, USER PROFILE, ITEM HISTROY ans ITEM PROFILE. Specifically, the user's strong preference for *funk, soul, and progressive rock* directly resonates with the item's characterization as a *funk-rock* fusion album. Similarly, the user's appreciation of *technical mastery and genre-defining innovation* aligns with the album's emphasis on *polished production, genre-blurring creativity, and tight musicianship*. This multi-level correspondence illustrates how DUET profiles condense and align user preferences with item characteristics. Such alignment provides the model with complementary signals beyond sparse histories, enabling more accurate predictions.

## 5 CONCLUSION

In this paper, we present DUET , a closed-loop framework for exploring and generating user–item textual profiles for recommendation. It comprises three key components: (1) *Cue Extraction*, which uses LLMs to distill user and item metadata into cues (serving as starting points for profile exploration); (2) *Self-Prompt Construction*, where the LLM iteratively rewrites and expands cues into candidate profiles; and (3) *Feedback-Driven Optimization*, which jointly optimizes user and item profiles via GRPO-based feedback-driven training. This design enables our framework to generate adaptive, high-quality profiles while retaining flexibility in profile representation. Experimental results across multiple datasets show that our method significantly outperforms existing baselines, validating the value of flexible, adaptive user profiles in enhancing recommendation performance.

## 6 LIMITATION

While utilizing reward signal from downstream recommendation performance effectively mitigates the risks of overfitting on static datasets and reward hacking, the training process is still complex and computationally intensive. Future work could focus on developing LLM acceleration algorithms specifically designed for LLM-based recommendation systems with real-time feedback.

ETHICS STATEMENT

This work does not involve human subjects, personally identifiable information, or sensitive data. The datasets used are publicly available, and all experiments comply with the ICLR Code of Ethics.

REPRODUCIBILITY STATEMENT

We have made extensive efforts to ensure reproducibility. The detailed methodology of our proposed approach is presented in Section 3, while the experimental settings, including training procedures and evaluation protocols, are described in Section 4. To further support reproducibility, we have made the complete source code and detailed instructions publicly available at https://duetreview.github.io/.

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

# A EXPERIMENT SETUP

## A.1 DATA CURATION

Table 4: Statistical details of the evaluation datasets.

| Dataset | #Train | #Valid | #Test | #User | #Item |
|---------|--------|--------|-------|-------|-------|
| Music | 43,071 | 3,271 | 1,296 | 4,183 | 2,660 |
| Book | 71,972 | 6,144 | 5,541 | 13,863 | 13,515 |
| Yelp | 51,497 | 4,757 | 4,328 | 8,453 | 13,426 |

We conduct experiments on three widely used real-world datasets:

- **Amazon Music (Music)**: This refers to the "Digital Music" subset of the well-known Amazon Product dataset[3], which records rich user reviews, ratings, and textual information about items, such as titles, across a broad range of product categories, on the Amazon platform.2

- **Amazon Book (Book)**: This refers to the "Book" subset of the Amazon Product dataset.

- **Yelp:** This refers to the Yelp Open dataset[4], which includes user reviews, ratings for businesses such as restaurants and retail shops, as well as textual information about the businesses. It is widely used in recommendation tasks Qiu et al. (2021).

We use the entire Music dataset for experiments, while for the Book and Yelp datasets, we utilize only a subset due to their large size. For the Book dataset, we use data from the last two months, and for the Yelp dataset, we use data from the last six months. For each dataset, we split it into training, validation, and test sets based on the timestamps of interactions, ensuring that test interactions occur after all training and validation interactions to prevent information leakage Ji et al. (2023).

Regarding data filtering, following prior work Liu et al. (2019), we adopt a 5-core setting to filter the data and exclude cold-start users and items—those not appearing in the training set—from the validation and test sets. The statistical details of the processed dataset are provided in Table 4.

### A.1.1 IMPLEMENTATION DETAILS

In our experiments, we primarily employ Qwen3-8B Team (2025) and LLaMA3 8B Instruct Dubey et al. (2024); Touvron et al. (2023) as both the recommendation model and the profile generation model. The training process is implemented using the TRL von Werra et al.. Key hyperparameters, such as batch size and learning rate, are determined through grid search to achieve optimal performance. More details can be found in our code.

## A.2 BASELINE PROMPTS

---
**KAR Prompt**

**Task:** Analyze user preferences based on business reviewing history
**Input:** {user_history} - User's business reviewing history with sentiments over time
**Instructions:**
1. Analyze the user's preferences considering business names and categories
2. Take into account sentiment patterns over time
3. Provide clear explanations based on reviewing history details
4. Consider other pertinent factors that may influence preferences

---

[3]https://cseweb.ucsd.edu/~jmcauley/datasets/amazon/links.html.
[4]https://business.yelp.com/data/resources/open-dataset/.

## PALR Prompt

**Task:** Summarize user preferences using keywords
**Input:** {user_history} - History businesses with keywords and user sentiments
**Output Format:** Itemized list based on importance
**Template:**
{1.KEY_WORD_1:"HISTORY_BUSINESS_1","HISTORY_BUSINESS_2";
  2.KEY_WORD_2:"HISTORY_BUSINESS_3"}
**Instructions:**
1. Extract key preference indicators from user interaction history
2. Rank keywords by importance
3. Associate relevant businesses with each keyword

## RLMRec Prompt

**Role:** Business recommendation assistant
**Task:** Determine business types a user is likely to enjoy
**Input Format:**
• Title: Business name
• Categories: Business categories
• Sentiment: User sentiment toward business

**Output Requirements:**
1. JSON format only
2. Structure:
{
  "summarization": "Types of businesses user likely enjoys" ($\leq$100 words),
  "reasoning": "Brief explanation for summarization" (no word limit)
}
3. No additional text outside JSON

**Input:** INTERACTION ITEMS: {user_history}

## LG Prompt

You will serve as an assistant to help me generate a user profile based on this user's sentiments history to better understand this users' interest and thus predict his/her sentiment about a target item. I will provide you with some behavior history of the user in this format: [item attributes and sentiment]. The user profile you generate should contain as much useful content as possible to help predict the user's sentiment towards a new business.
**USER HISTORY:** [user history].
**PROFILE YOU GENERATE:**

## R4Rec Prompt (Reasoner)

**User Review History**
$\langle H_u$ organized as below$\rangle$
1. Title of Item 1
Positive Aspects: [Aspect 1], [Aspect 2], ...
Negative Aspects: [Aspect 1], [Aspect 2], ...
User Preference Elements: [Preference 1], [Preference 2], ...
2. Title of Item 2
Positive Aspects: [Aspect 1], [Aspect 2], ...
Negative Aspects: [Aspect 1], [Aspect 2], ...
User Preference Elements: [Preference 1], [Preference 2], ...
...

**Item Review History by Other Users**
$\langle H_i$ organized in the same format as above$\rangle$
Task: Analyze whether the user will like the new Music $i$ based on the user's preferences and the item's features. Provide your rationale in one concise paragraph.

## B THE USE OF LARGE LANGUAGE MODELS

We used a Large Language Model (LLM) only as a writing assistant to polish the language of the manuscript (*e.g.*, grammar refinement, style adjustment, and clarity improvement). The research ideas, methodology design, experiments, and analysis were entirely conceived, implemented, and validated by the authors without reliance on the LLM. The LLM did not contribute to research ideation, experimental design, or result interpretation.

