# OpenReview forum: "Duet: Joint Exploration of User–Item Profiles"
_ICLR.cc/2026/Conference — ICLR 2026 Conference Withdrawn Submission_

### Official Review · Reviewer_hRi7 · 2025-10-30

**Soundness:** 2
**Presentation:** 3
**Contribution:** 3
**Rating:** 6
**Confidence:** 3

**Summary:**

This paper introduces DUET, a framework designed to shift recommendation systems (RS) from traditional latent vector representations to interpretable, textual profiles for both users and items. The central problem it addresses is that the optimal format for these textual profiles is unknown, and handcrafting templates is often ineffective and misaligned with the final recommendation task.

Three stages are needed:

1. A Large Language Model (LLM) first distills raw user histories and item metadata into concise, informative textual "cues" that act as starting points for profile generation
2. In a single sequence-to-sequence pass, the framework expands these minimal cues into richer, more descriptive textual profiles. This unique step allows the model to explore various potential profile formats rather than being restricted to a single, predefined structure.
3. the generated user and item profiles are then optimized together using reinforcement learning.

Key contributions:

* a new paradigm for RS. moving from latent vectors to align users and items using natural language profiles in a shared semantic space. A innovative move to enable both human interpretation and model LLM/agentic system.

* exploration is a drive force for recommendation. the work here introduces a novel method that empowers an LLM to autonomously discover effective profile formats without relying on rigid, hand-engineered templates. This is achieved through the cue-initialization and self-prompting mechanism.

* empirical evaluation results show that DUET significantly outperforms strong existing baselines, confirming the effectiveness of both the textual alignment approach and the feedback-driven profile exploration strategy

**Strengths:**

originality:

* vector to text: The core idea of shifting from opaque latent vectors to interpretable textual profiles for both users and items is a significant contribution. I am not quite familiar with a comprehensive understanding of existing work though. To me, this joint modeling seems novel.

* The proposed three-stage DUET framework is a novel combination of techniques. The concept of "self-prompt construction" to allow the LLM to autonomously explore and discover optimal profile formats—rather than relying on rigid, handcrafted templates—is a particularly creative and original mechanism

* The use of RL to create a closed-loop system where downstream recommendation performance directly provides the reward signal for refining profiles is an elegant and powerful idea. This approach moves beyond offline reward models and directly optimizes for the end-task, addressing a key limitation in prior RL-based RS

Quality:

*  The proposed method is well-designed and technically sound.
* Evaluation seems valid to me including the ablation study, though I am not that familiar with the baseline method or any of the state-of-art work.

Clarity:
The paper is clearly written, easy to follow.

Significance
the paper offers a forward-looking vision for the future of RS in the era of large language models.

* as I mentioned in the summary, it provides a foundation for agentic systems
* address a key LLM challenge: designing effective prompts and profile formats
* The demonstrated gains in accuracy and F1-score are significant enough to be of interest to both researchers and practitioners in the field.

**Weaknesses:**

* The experiments exclusively use rating prediction metrics (MAE, RMSE, Accuracy, F1) to evaluate performance. While this is a valid approach, modern recommendation systems are fundamentally ranking problems. The current evaluation does not assess how well the generated profiles perform in a more realistic scenario of ranking a large set of candidate items. Actionable Insight: The work would be significantly strengthened by including experiments on a re-ranking or full candidate retrieval task, using standard ranking metrics like nDCG, MAP, and Recall@K. This would provide more direct evidence of the profiles' effectiveness in a real-world setting.

* as the author already pointed out, the computational need would be super high.

* The use of RL (specifically GRPO) is central to the framework's success, but RL for LLMs can be notoriously unstable and sensitive to hyperparameter choices. The paper does not discuss the stability of the training process or the sensitivity to RL-specific hyperparameters. The paper would be more robust if it included an analysis of the RL training dynamics. For instance, showing the learning curves for reward and providing details on hyperparameter tuning would increase confidence in the method's reproducibility and stability. Comparing GRPO to another common policy optimization algorithm (like PPO) could also demonstrate that the gains are from the framework itself and not just the choice of a specific, state-of-the-art RL algorithm.

* Certain aspects of the methodology, particularly the novel components, could benefit from more detailed explanation.
e.g.,  The initial "cue-based initialization" stage is described at a high level. The prompt guiding the LLM to extract cues is quite broad (e.g., "Keep the description concise and avoid full sentences"). The quality and nature of these initial cues seem critical to the success of the subsequent stages, yet this is not analyzed.  The paper could be improved by adding a qualitative analysis of the generated cues. Furthermore, a sensitivity analysis showing how the final profile quality is affected by variations or perturbations in the initial cues would provide a better understanding of the method's robustness.

**Questions:**

* Is the success of the joint optimization stage fundamentally tied to GPRO, or is the framework general enough to work with other policy optimization methods like PPO? A response here would help clarify whether the core contribution is the closed-loop framework itself or the application of a specific state-of-the-art RL algorithm.

* The case study in Figure 4 provides a strong example of success. However, could you discuss potential failure modes of DUET? For instance, how does the framework handle users with very sparse histories, or users whose historical interactions contain conflicting or rapidly changing preferences? Understanding these limitations would provide a more complete picture of the framework's applicability.

* The paper motivates its work by stating that textual profiles establish a foundation for "future agentic recommendation systems". This is a very compelling vision. Could you please elaborate on what a downstream agentic system powered by DUET profiles might look like? A more concrete example would help solidify the long-term significance and impact of the proposed work.

---

### Official Review · Reviewer_sG5X · 2025-10-30

**Soundness:** 2
**Presentation:** 2
**Contribution:** 2
**Rating:** 2
**Confidence:** 3

**Summary:**

The paper introduces DUET, a closed-loop framework leveraging Large Language Models (LLMs) for profile generation in recommendation task. DUET includes three main steps: i) prompting LLMs to generate a short phase capturing minimal yet informative user interests/item characteristic; ii) re-prompting LLMs to expand cues into a richer user/item profile; iii) aligns generated user and item profiles for rating prediction task using reinforcement learning. Experiments on three real-world datasets show the stronger rating prediction performance of DUET than representative baselines.

**Strengths:**

1. Motivation: The paper presents a well-motivated study that explores the use of large language models (LLMs) for informative profile generation. This approach effectively leverages the world knowledge embedded in LLMs to enrich user and item representations beyond traditional sparse interaction data.

2. Organization and Clarity: The paper is well structured and clearly written. The logical flow of sections facilitates comprehension, and the inclusion of concrete prompt examples enhances the reader’s understanding of the core methodology

3. Experimental Results: The proposed model, DUET, demonstrates substantial improvements in rating prediction accuracy compared to representative baselines. These results validate the effectiveness of the proposed profile generation approach.

**Weaknesses:**

1. While the motivation and empirical results are promising, the paper does not clearly articulate how the generated user and item profiles concretely advance the recommendation task. A more detailed analysis is needed to explain why these profiles lead to improved predictions, e.g., case studies, ablation experiments on specific prompt designs, or comparisons between profiles generated by DUET and those from baselines to highlight the unique advantages of the proposed approach.

2. The proposed cue-based mechanism for profile generation is conceptually interesting but not fully convincing. Since cues are often short phrases containing limited information, the resulting profiles may not faithfully reflect the true user preferences. For example, users sharing a similar cue might still have distinct underlying interests. The paper would benefit from a more in-depth discussion or empirical validation showing how DUET mitigates this issue and ensures that generated profiles remain representative and reliable.

3. It remains unclear why the paper focuses exclusively on the rating prediction task rather than ranking-based evaluation, which is typically more aligned with the goals of recommender systems: identifying and retrieving relevant items for users. Prior work (e.g., [a]) has highlighted the limitations of rating prediction for assessing recommender effectiveness. Furthermore, many of the chosen baselines were originally designed for ranking tasks, which makes the comparison less meaningful in the current setup. Including ranking-based evaluations would substantially strengthen the empirical validation and demonstrate the broader applicability of DUET.

[a] Cremonesi, P., Koren, Y., & Turrin, R. Performance of recommender algorithms on top-N recommendation tasks. RecSys 2010.

4. The paper acknowledges that the proposed method introduces additional complexity and computational overhead. However, no quantitative analysis is provided to assess the trade-off between performance gains and computational cost. A detailed efficiency study would provide valuable insight into the practicality of deploying DUET in real-world scenarios.

**Questions:**

Please see the review.

---

### Official Review · Reviewer_3Gyu · 2025-11-01

**Soundness:** 2
**Presentation:** 2
**Contribution:** 1
**Rating:** 4
**Confidence:** 3

**Summary:**

This paper presents DUET, a framework for making textual user/item profiles for LLM-based recommenders. Instead of using fixed templates, which often don't work well, DUET learns the profiles. It starts by making short "cues" from raw data, expands those cues into full profiles, and then uses RL (with GRPO) to jointly optimize both user and item profiles based on how well they work for the actual recommendation task. Experiments on three datasets show this approach beats strong baselines.

**Strengths:**

The best part is moving beyond static templates. The "cue" to "profile" idea is a smart way to get around the pain of prompt engineering.

Another big strength is optimizing both user and item profiles together. Most work only focuses on the user. Using RL to align both based on task performance seems to be the right way to go and helps capture better user-item matches (like in Fig 4).

The technical details, like using GRPO and the fractional reward, are well-thought-out for applying RL to this problem. The strong results and clear ablation study really sell the idea.

**Weaknesses:**

My main worry is the computational cost. This looks very expensive. It needs LLM passes for profile generation and a full RL optimization loop. The paper mentions this in the limitations but doesn't give any analysis of training time or inference latency vs. the baselines. This is a big practical issue.

Also, the "exploration" part of the profile construction isn't very clear. Figure 3 makes it look like a single-pass generation (data -> cue -> prompt -> profile). How much "exploration" is really happening? Is it just a fixed refinement?

**Questions:**

Can you give us some idea of the computational cost? How much slower is DUET to train and run compared to the baselines? This is a key practical concern.

Can you clarify what "exploration" means in the self-prompt construction step? Is it just a single-pass refinement, or is the model actually trying out different profile formats (e.g., through sampling)?

For Table 1, did the baseline models (KAR, PALR) also use generated item profiles? Or just user profiles? This is key for a fair comparison of the "joint" optimization.

---

### Official Review · Reviewer_vEKn · 2025-11-01

**Soundness:** 1
**Presentation:** 3
**Contribution:** 2
**Rating:** 2
**Confidence:** 4

**Summary:**

This study proposes a framework that generates user and item textual profiles jointly and uses these profiles for the recommendation task. Experimental results show that the proposed method outperforms some prompting-based baselines.

**Strengths:**

(1) Writing is easy to follow.

**Weaknesses:**

(1) It would be better if the authors could further discuss results in Table 1. Table 1 shows that the proposed method significantly outperforms other baselines. However, I wonder if the superior performance is related to the RL phase where the model is optimized while other baselines are simply prompting-only methods. In fact, by checking the numbers in both Table 1 and Table 2, it seems that RL is the major factor that leads to the best performance. By adding Profile and Self-Prompt, the proposed method does not perform as well as other baselines.

(2) More quantitative experiments are necessary to support the discussion in Section 4.3.2. The authors discussed three potential reasons for the inconsistent performance of the proposed method across different user interaction lengths without providing quantitative support. Moreover, is the performance difference significantly large to make any conclusions?

(3) It would be better if the authors can improve the case study and expand it into a large scale quantitative experiments. It is natural that there are some good examples demonstrating the advantage of the proposed method. It is also not surprising if some similar examples can be found in the responses generated by other baselines. But to articulate the advantage of the proposed method, it is necessary to show that such advantage (1) is indeed the reason for the better performance, and (2) appears in the responses of the proposed method significantly more commonly than in the responses of other baselines.

**Questions:**

(1) The authors mentioned in Appendix A.1.1 that hyperparameter details can be found in the code, but no code was shared. Can the authors provide more details about how the framework was trained?

(2) I wonder if the authors have considered the cold-start issue? How would the proposed framework address this issue?

---

### Note · Authors · 2026-01-06

I have read and agree with the venue's withdrawal policy on behalf of myself and my co-authors.